# *Ex Vivo* Murine Skin Model for *B. burgdorferi* Biofilm

**DOI:** 10.3390/antibiotics9090528

**Published:** 2020-08-19

**Authors:** Jason P. Torres, Alireza G. Senejani, Gauri Gaur, Michael Oldakowski, Krithika Murali, Eva Sapi

**Affiliations:** Department of Biology and Environmental Science, University of New Haven, 300 Boston Post Road, West Haven, CT 06516, USA; jaypaultorres@gmail.com (J.P.T.); asenejani@newhaven.edu (A.G.S.); ggaur2@unh.newhaven.edu (G.G.); molda1@unh.newhaven.edu (M.O.); kmura4@unh.newhaven.edu (K.M.)

**Keywords:** Lyme disease, antibiotics resistance, biofilm, confocal, atomic force microscopy

## Abstract

*Borrelia burgdorferi*, the causative agent of Lyme disease, has been recently shown to form biofilm structures in vitro and *in vivo*. Biofilms are tightly clustered microbes characterized as resistant aggregations that allow bacteria to withstand harsh environmental conditions, including the administration of antibiotics. Novel antibiotic combinations have recently been identified for *B. burgdorferi in vitro*, however, due to prohibiting costs, those agents have not been tested in an environment that can mimic the host tissue. Therefore, researchers cannot evaluate their true effectiveness against *B. burgdorferi*, especially its biofilm form. A skin ex vivo model system could be ideal for these types of experiments due to its cost effectiveness, reproducibility, and ability to investigate host–microbial interactions. Therefore, the main goal of this study was the establishment of a novel ex vivo murine skin biopsy model for *B. burgdorferi* biofilm research. Murine skin biopsies were inoculated with *B. burgdorferi* at various concentrations and cultured in different culture media. Two weeks post-infection, murine skin biopsies were analyzed utilizing immunohistochemical (IHC), reverse transcription PCR (RT-PCR), and various microscopy methods to determine *B. burgdorferi* presence and forms adopted as well as whether it remained live in the skin tissue explants. Our results showed that murine skin biopsies inoculated with 1 × 10^7^ cells of *B. burgdorferi* and cultured in BSK-H + 6% rabbit serum media for two weeks yielded not just significant amounts of live *B. burgdorferi* spirochetes but biofilm forms as well. IHC combined with confocal and atomic force microscopy techniques identified specific biofilm markers and spatial distribution of *B. burgdorferi* aggregates in the infected skin tissues, confirming that they are indeed biofilms. In the future, this ex vivo skin model can be used to study development and antibiotic susceptibility of *B. burgdorferi* biofilms in efforts to treat Lyme disease effectively.

## 1. Introduction

Lyme disease is a vector-borne illness that is caused by *B. burgdorferi* sensu lato, a bacterial spirochete that is transmitted by Ixodes ticks [1]. After the tick bite, *B. burgdorferi* disseminates in the skin, and the most common manifestation of infection is a red rash called erythema migrans [2]. The other well-studied dermatological conditions of Lyme disease are Borrelial lymphocytoma (BL), which appears in the early phase of Borrelia infection, and acrodermatitis chronica atrophicans (ACA), which is the late onset cutaneous manifestation [3,4]. However, Lyme disease is a multi-systemic disease, and patients could experience severe chronic health conditions such as Lyme carditis, arthritis, and neuroborreliosis [5,6,7]. Frontline treatments for early Lyme disease cases involve using antibiotics such as doxycycline, amoxicillin, cefuroxime, and ceftriaxone [7,8,9,10,11]. Unfortunately, many studies have suggested that, in late stages of Lyme diseases, *B. burgdorferi* can persist in the body following the antibiotic treatment due to dissemination into different organs [12,13,14,15,16,17]. These findings have further been validated extensively through studies in mice, dog, and rhesus macaques following various antibiotic treatments [18,19,20,21,22,23].

Several possibilities for the persistent symptoms have been suggested after the discontinuation of antimicrobial therapies [24,25,26,27], including the development of antibiotic resistant alternative forms [28]. It was well documented that, in addition to the spirochetal form, *B. burgdorferi* can adopt other various morphological forms such as round bodies/cysts and biofilms in response to unfavorable environmental conditions such as antibiotic therapy [29,30,31,32,33,34,35,36,37]. *B. burgdorferi* biofilm forms have been shown to greatly persist following antibiotic treatment in vitro [38]. This resistant form of *B. burgdorferi* has been found first in vivo in Borrelial lymphocytoma biopsy tissues [39]. We provided further evidence that Borrelia biofilm form can also be found in other infected human organs [40]. In our previous study, human autopsy tissues from a well-documented serum, PCR, and a culture positive Lyme disease patient who died despite extensive antibiotic treatments over the course of her 16 year-long illness were examined. Tissues from brain, heart, kidney, and liver not just revealed significant pathological changes but also demonstrated the presence of Borrelia spirochetes and biofilms with inflammatory markers [40].

Biofilms are aggregations of microorganisms that produce a protective extracellular polymeric matrix mainly consisting of polysaccharides such as alginate in addition to proteins, lipids, and extracellular DNA [41,42]. These components all play crucial roles not just in overall protection but in adhesion and structural integrity of the biofilm as well [43,44,45]. In the human body, biofilm formation by pathogenic species is one of the main causes of developing chronic diseases [46]. It was estimated that biofilm form can provide a 1000-fold resistance to our current antimicrobial agents [47]. The highly antibiotic resistant nature of pathogenic biofilm is driving a major clinical concern, and many efforts have been made to develop in vitro and in vivo model systems that can study biofilm formation and its antibiotic resistance with the goal to mimic the natural environment where infection occurs [48]. While the in vitro models offer low cost, easy set-up, and potential high throughput screening, it does not assess environmental factors and the host–microbial interactions accurately [49].

There is an urgent need to develop a model system that can help the development and the antibiotic resistance of the biofilm structures in a sustainable biological environment in order to further assess host–microbial interactions.

*Ex Vivo* models are “out of the body” systems that are derived from a living organism and are kept alive in an external environment for research purposes [50]. *Ex Vivo* systems are advantageous as infection models, as they allow researchers to closely monitor and evaluate host–microbial interactions [51,52]. In addition, ex vivo systems are cost efficient, highly reproducible, and easier to maintain in comparison to animal models [52]. They are also ideal systems to image or analyze the progression of bacterial colonization and the biofilm development in a specific tissue [53]. An example of a developed ex vivo system was the ex vivo porcine lung model that was generated in order to study growth, virulence, and signaling of *Pseudomonas aeruginosa* [51]. In addition, novel ex vivo skin infection models have been generated in order to combat antibiotic resistance behaviors of emerging bacterial strains [52]. Previous studies suggested an excessive induction of pro-inflammatory cytokines after an infection of *Staphylococcus aureus* and *P. aeruginosa* in a novel ex vivo human skin chamber model [51,52,53].

*Ex Vivo* models were already developed for *B. burgdorferi* spirochetal research in human skin, tonsillar, and Rhesus brain tissues [54,55,56] with the propose of exploring the molecular mechanism of invasion and host gene activation after *B. burgdorferi* infection. However, biofilms in this ex vivo tissue models were not established nor studied.

Therefore, the objective of this study was to develop an ex vivo murine skin model that is suitable for borrelial biofilm research. *B. burgdorferi* spirochetes were inoculated into murine skin samples at various concentrations, cultured in different media and archived infected biopsies were generated. These sections were then analyzed utilizing immunohistochemistry (IHC) with *B. burgdorferi* and alginate-specific (biofilm marker) antibodies for the fluorescent detection of these targets. Once *B. burgdorferi* spirochetes and biofilms were identified, a reverse-transcriptase PCR protocol was performed to evaluate if *B. burgdorferi* remained alive in the skin biopsies two weeks post-inoculation. Furthermore, confocal microscopy method was performed in order to analyze the integration of *B. burgdorferi* spirochetes and biofilm in the ex vivo skin biopsies. Finally, atomic force microscopy was used to further analyze these structures for additional biofilm characteristics. In summary, this research offers a novel ex vivo murine skin model for *B. burgdorferi* research. This system can be heavily utilized in the evaluation of antibiotic resistance of *B. burgdorferi*.

## 2. Results

### 2.1. Establishment of an Ex Vivo Skin Model System for a B. burgdorferi Biofilm

Several culture conditions were tested to evaluate their effect on the morphology of *B. burgdorferi* in the murine skin tissues. Those conditions were selected based on experimental data from previously published ex vivo models for *B. burgdorferi* [54,55,56]. First, different concentrations of spirochetes were injected into murine skin punch biopsy samples. The infected tissue samples were then cultured in either BSK-H supplemented with 6% rabbit serum (RS), the preferred growth media for *B. burgdorferi*, or DMEM supplemented with 10% calf serum (CS), the preferred growth media for mammalian cells, for 14 days.

### 2.2. Analyses of Infected Murine Skin Biopsies Inoculated with 5 × 10^6^ Spirochetes

In the first set of experiments, murine skin biopsies were inoculated with a total concentration of 5 × 10^6^ spirochetes and cultured in either BSK-H + 6% RS or DMEM + 10% CS media for 14 days (2 × 30 samples). IHC staining was performed utilizing an anti-*B. burgdorferi* and anti-alginate antibody (a biofilm marker). The IHC findings demonstrated the presence of *B. burgdorferi* spirochete in tissues cultured in either media (green staining, Figure 1A,F) in all infected skin biopsy samples. *B. burgdorferi* positive spirochetes did not, however, stain positive for alginate biofilm marker (Figure 1B,G), suggesting that they did not produce biofilm form. *B. burgdorferi* spirochetes were quantified in 30 positive slides/each culture condition representing individual sections of the 30 skin biopsies that were inoculated with 5 × 10^6^ spirochetes and cultured in either BSK-H + 6% RS or DMEM + 10% CS culture media. Results showed that all tissues sections from both culture conditions had more than 500 spirochetes/mm^2^, but no biofilm structures were found regardless of the media types used (Table 1 and Table 2). Uninfected murine skin biopsies from the same animal were used as negative controls (30 samples), and the obtained data showed that there was no *B. burgdorferi* or alginate positive staining observed in any of those sections (Figure 1K,L). An additional negative control, non-specific IgG antibody, was used instead of the primary antibodies in parallel IHC experiments, which resulted in no positive staining on any of the infected murine skin sections (Figure 1C,H,M).

### 2.3. Analyses of Infected Murine Skin Biopsies Inoculated with 1 × 10^7^ Spirochetes

In parallel experiments, another set of murine skin biopsies (2 × 30 tissues) were inoculated with total concentration of 1 × 10^7^ spirochetes and cultured in either BSK-H + 6% RS or DMEM + 10% CS media for 14 days, and the obtained results were quantified on individual sections from all infected tissues using IHC staining for *B. burgdorferi* and alginate combined with fluorescent microcopy.

Results shows that large amounts of *B. burgdorferi* spirochetes were found in every section studied, regardless of which media was used, and all of the sections had more than 500 spirochetes/mm^2^ (Table 1). In addition to the identification of the spirochetal morphological form of *B. burgdorferi* in skin biopsies inoculated with 1 × 10^7^ cells, there were *B. burgdorferi* and alginate positive aggregates, indicating the presence of *B. burgdorferi* biofilm inside the skin explants cultured in either BSK-H + 6% RS or DMEM + 10% CS media (Table 2).

Figure 2 is a representative image of the IHC findings. *B. burgdorferi* positive aggregates (green staining Figure 2A,F) also stained positive for the biofilm marker alginate (Figure 2B,G) when cultured in either BSK-H + 6% RS or DMEM + 10% CS media. *B. burgdorferi* positive spirochetes (green staining, Figure 2K) did not stain with alginate antibody (Figure 2L), indicating they were not in biofilm form. There were two negative controls included in these experiments: IHC staining on sections from uninfected murine skin biopsies and the use of non-specific IgG antibody instead of primary antibody. None of the negative controls resulted in any *B. burgdorferi* or alginate positive staining (Figure 2P,Q,R and Figure 2C,H,R, respectively).

Table 2 summarizes the quantitative analyses of the obtained biofilm structures in the different cultures. There was no biofilm formation found in any of the infected biopsies when it was inoculated with 5 × 10^6^ spirochetes and cultured for 14 days in either BSK-H 6% RS media or DMEM + 10% CS media. On the other hand, there were 12/30 slides that had 1–2 biofilms/slide in the skin biopsies that were inoculated with 1 × 10^7^ spirochetes and cultured in BSK-H 6% RS media for 14 days (Table 2). In addition, there were 8/30 slides that had one biofilm/slide in skin biopsies that were inoculated with 1 × 10^7^ spirochetes and cultured in DMEM + 10% CS media for 14 days. (Table 2). Statistical analyses, however, showed no significant differences between the number of obtained biofilms from the two culture methods with different type of media (*p* value > 0.05).

### 2.4. Reverse-Transcriptase PCR (RT-PCR) Analysis on Infected Murine Skin Biopsies

Since there were more *B. burgdorferi* biofilms detected in the murine skin biopsies that were inoculated with 1 × 10^7^ spirochetes and cultured in BSK-H media supplemented with 6% rabbit serum (Table 2), RNAs from 10 frozen tissue samples were extracted and used as templates for the RT-PCR experiments in order to determine if *B. burgdorferi* remained live two weeks post-infection. Complementary DNA (cDNA) was made from the extracted RNA templates, and *B. burgdorferi* specific PCR was performed (see Material and Methods). The PCR products were analyzed by standard agarose gel-electrophoresis and direct sequencing methods. Figure 3 shows a representative agarose gel image of the PCR experiments. Positive control for this experiment consisted of extracting RNA from *B. burgdorferi* B31 strain culture directly, which yielded the 450-base pair band, the expected size of the 16S ribosomal DNA target sequence (Lane 3). Infected murine skin sample yielded a similar band of 450 base pairs (Lane 4). Negative controls included *Escherichia coli* culture cDNA (*E. coli*, Lane 2), infected murine skin with no cDNA template (Lane 5), infected murine skin sample treated with RNase prior to the RT-PCR step (Lane 6), and uninfected murine skin cDNA (Lane 7), all of which used the same 16S rDNA-based PCR protocols. None of the negative controls yielded any visible PCR bands. The obtained positive PCR bands from the *B. burgdorferi* B31 strain culture and from the ex vivo experiments were purified and sequenced as described in Material and Methods. Basic Local Alignment Search Tool (BLAST, NBCI) analyses confirmed a 100% identity to *B. burgdorferi* B31 strains in all of those samples (data not shown).

### 2.5. Confocal Microscopy Analysis of Various Morphological Forms of B. burgdorferi in Infected Murine Skin Biopsies

Confocal microscopy analyses were performed on skin biopsies infected with 1 × 10^7^ spirochetes and cultured in BSK-H + 6% RS medium via IHC methods. The samples were first co-stained with a *B. burgdorferi*-specific and an alginate-specific antibody (biofilm marker) and scanned using a confocal microscope. Figure 4 shows an aerial view of individual channels for alginate (red staining, Figure 4A) and for *B. burgdorferi* (green staining, Figure 4B) fluorescent images. Merged channels were also constructed to visualize the spatial distribution of these alginate and Borrelia markers (Figure 4C). Figure 4D shows an additional combined image with differential interference contrast microscopy, alginate, and *B. burgdorferi* IHC staining to demonstrate that *B. burgdorferi* biofilm structures were truly embedded in the mammalian tissue.

To further visualize the spatial distributions of the biofilm structures, a 3D confocal microscopy scanning was performed combining individual Z stacks.

Appendix A demonstrates a different view of the Figure 4 IHC images with a 3D construction of the individual confocal microscopy channels representing alginate (Figure 4A, red staining), *B. burgdorferi* (Figure 4B, green staining), DAPI (blue staining), and differential interference microscopy (DIC) microscopy image (Figure 4C) to further illustrate the spatial distribution of the biofilm structures. The Z-stack analyses indicated that this particular biofilm was about 120 μm × 100 μm × 3 μm in dimension.

To further analyze the spatial distribution of the *Borrelia* biofilm with alginate staining, confocal microscopy analyses of the same tissue section performed using individual z-stacks to form a composite 3D image (using Image J program). Figure 5 shows a 3D construction of the merged z-stacked images from three confocal microscopy channels with alginate (red staining), *B. burgdorferi* (green staining), and DAPI (blue staining).

### 2.6. Atomic Force Microscopy Analysis of B. burgdorferi Biofilm in Infected Murine Skin Biopsies

In order to assess the structural organization of the *B. burgdorferi* biofilm in murine skin biopsies infected with 1 × 10^7^ spirochetes and cultured in BSK-H + 6% RS medium for 14 days, 3D topography of *B. burgdorferi* biofilms embedded in ex vivo murine skin biopsies was captured utilizing a contact-mode atomic force microscope as previously described [36,37]. Figure 6 demonstrates that biofilms formed by *B. burgdorferi* in the infected murine skin ex vivo tissue had the characteristic “tower morphology” of *B. burgdorferi*, as previously demonstrated in Borrelial lymphocyte human biopsy tissues [39]. The skin tissue-embedded ex vivo *B. burgdorferi* biofilms had channels and protrusions (Figure 6, red and green arrows, respectively) that are hallmark characteristics of biofilms formed in Borrelial lymphocytoma human skin tissues [39]. In good agreement with the confocal analyses, the depth of the biofilm was about 3–5 μm.

## 3. Discussion

The purpose of this study was to establish an ex vivo murine skin model system to examine *B. burgdorferi* biofilm formation. The obtained results showed that biofilm structures can be developed by injecting 1 × 10^7^ spirochetes into murine skin punch biopsies and culturing them in BSK-H media supplemented with 6% RS or DMEM media supplemented with 10% CS for 14 days.

In previous ex vivo model systems for *B. burgdorferi* research, investigators have chosen human tonsillar, skin, and Rhesus brain tissues [54,55,56]. In this study, murine skin biopsies were chosen for several reasons. Skin is the primary site to *B. burgdorferi* infection after a tick bite, and it is well known that *B. burgdorferi* can establish persistent infection in mice skin [57]. Furthermore, murine skin biopsies have a great advantage in that they are easily available from surplus leftover material from various rodent research studies.

One of the key questions of this study was whether *B. burgdorferi* cells stay alive after two weeks of culturing in the skin tissues. To answer this question, frozen tissue sections from these infected biopsies were analyzed for the presence of Borrelia RNA using RT-PCR methods. The result from RT-PCR study showed active transcription was indeed taking place in *B. burgdorferi* cells. However, because the tissues contained large number of spirochetes, which were not associated with biofilm structures, we could not determine the exact origin of the active gene expression. The other important question was whether the observed *B. burgdorferi* biofilm-like aggregates are indeed biofilm structures. If so, it was expected to possess biofilm characteristics that were first identified in vitro in cultures and in vivo in Borrelial lymphocytoma tissues [39]. We provided several lines of evidence to confirm that they are true biofilm structures, including (1) IHC analyses combined with confocal microscopy revealed the active expression of alginate in those aggregates. Alginate was previously shown to be a major mucopolysaccharide of the extracellular matrix protecting *B. burgdorferi* biofilm structures [36,37,39,40]; (2) merged confocal images of *B. burgdorferi*, alginate (biofilm marker), and differential interference contrast microscopy showed that the *B. burgdorferi* aggregates were seeded deep in the skin tissues; (3) atomic force microcopy analyses demonstrated another classical biofilm feature called “the tower morphology” of the *B. burgdorferi* aggregates with channels and protrusions, resembling what has been previously demonstrated as *B. burgdorferi* biofilm structures in Borrelial lymphocytoma human biopsy tissues [39]. The observed *B. burgdorferi* biofilm channel-like structures were found in other biofilm forming species, such as *P. aeruginosa* [58], *Azotobacter vinelandii* [59], and *Leptospira biflexa* [60]. Those channels have a very important function for the survival of the bacterial community by providing routes for nutrition to enter and waste to exit [36,39,41,58]. The proof that *B. burgdorferi* biofilms are found in the ex vivo skin tissue explants embedded in the tissue, have biofilm channels, and express alginate, demonstrates that they indeed have the true characteristics of pathogenic biofilms.

The next question was if the sizes and the numbers of *B. burgdorferi* biofilm structures found in the ex vivo skin tissues represent the sizes and the number of biofilms found previously in vivo in skin and other infected human tissues [39,40]. Those previous studies revealed the size of *B. burgdorferi* biofilms can vary from 20–300 μm in human skin and other organs such as brain, heart, kidney, and liver [40]. The sizes of *B. burgdorferi* biofilms in ex vivo mouse skin explants studied here were in a very similar range sized from 20–300 μm. The heights of the biofilms found in the murine ex vivo explants were about 3–5 μm, which was about the size we found in the Borrelia biofilm positive human autopsy tissues [40]. The numbers of *B. burgdorferi* biofilms found in ex vivo skin tissues studied here also were in good agreement with the numbers of biofilms identified in human organs (one to two biofilm structures per tissue section), demonstrating that this ex vivo model closely resembles in vivo infection [39,40].

As stated above, there are several human ex vivo models (skin, tonsillar, brain) established previously for *B. burgdorferi* research, and they were successfully used to study inflammatory responses and other mechanisms of host–pathogen interactions [54,55,56]. Similarly, ex vivo human skin models were also established for *P. aeruginosa* and *S. aureus* [61,62] by inoculating bacteria at various concentrations into skin biopsies and analyzing different inflammatory pathways. While those studies provide an important insight of pathogenic infection of human tissues, they have some disadvantages. *Ex Vivo* human skin models are expensive, difficult to access, and, most importantly, they are well known to have inter-individual variables [63]. *Ex Vivo* murine skin models from specific mouse clones would overcome these issues and could provide a very useful model system to study *B. burgdorferi* infection. The other issues were that the above-mentioned study did not investigate the presence of alternative morphological forms of those pathogens, which was recently suggested to be a potential major factor of the different host responses [64]. For example, results of a recent study suggested that secreted products from *S. aureus* biofilms can affect the host response differently than *S. aureus* planktonic cell in a cultured keratinocyte model [64]. Interestingly, in this study, they found differences in various pathways such as in inflammatory, apoptotic, and nitric oxide responses as well IL-6, IL-8, TNFα, and CXCL2 productions between keratinocytes cells exposed to either *S. aureus* biofilm or planktonic cells [64].

The inflammatory responses for *B. burgdorferi* biofilms were recently studied in autopsy tissues of well documented serum-, PCR-, and a culture-positive Lyme disease patient who died after extensive antibiotic treatments over the course of her 16-year-long illness [40]. Immunohistochemistry analyses of several organs revealed a significant number of infiltrating CD3+ T lymphocytes present in the vicinity of *B. burgdorferi* biofilms but not next to the spirochetal forms [40]. These findings indicating the importance for a better understanding of how the biofilm form interacts with the host environment. Therefore, we need model systems that include not only planktonic cells but also biofilm aggregates.

The other possible further advantage of our novel ex vivo murine model is that it can be used to study multi-species biofilms, which are reported for several chronic conditions such as chronic wounds and dental plaques [65]. Those conditions are well known to be the most difficult infections to clear [66,67]. Multispecies biofilms were also recently discovered for *B. burgdorferi* [68,69]. In two different dermatological human diseases (Borrelial lymphocytoma and Morgellons diseases), *Chlamydia* spp. and *Helicobacter pylori* species were found respectively inside of the *B. burgdorferi* biofilm structures [68,69]. These findings strongly suggest that we need to establish multi-pathogen infection model systems for the further understanding of those complex infections. Obviously, the final goal is to develop effective antimicrobial therapeutic strategies to more broadly eliminate chronic infections.

Antibiotic and antimicrobial testing for various morphological forms of *B. burgdorferi*, including spirochetes, cyst/round body, and biofilm forms, is primarily conducted in vitro with novel as well as certain combinations of previously discovered antimicrobial agents [70,71,72,73,74,75,76,77,78,79]. Those studies provided strong evidence that the most antibiotic resistant form of *B. burgdorferi* is the biofilm [38,73,75,76], which agrees with the data for biofilm resistance of other pathogenic species [43,44,45,46,47]. To further confirm these important findings, a recent in vivo study demonstrated that inoculation of *B. burgdorferi* biofilm-like microcolonies into a mouse model for Lyme arthritis caused a more severe and antibiotic tolerant infection than inoculating the mice with individual spirochetes [80]. These results further suggest that we need a model system that allows the study of Borrelial biofilm persistence and resistance for antimicrobial agents and as one that mimics the host environment well.

In summary, this study describes the development of a novel ex vivo murine skin model for *B. burgdorferi* biofilm research. This model can be used to assess the susceptibility of these resistant structures to various antibiotics and antimicrobial agents. Furthermore, this would allow us to study the host responses to biofilm forms in efforts to develop new therapeutic approaches for Lyme disease treatment.

## 4. Material and Methods

### 4.1. Bacterial Culture

Low passage isolates (<6) of *B. burgdorferi* B31 strain (ATCC 35210, Manassas, VA, USA) were grown in BSK-H media (Sigma, St. Louis, MO, USA) supplemented with 6% rabbit serum (Pel-Freeze, Rogers, AR, USA) in 15 mL glass tubes at 33 °C and 5% CO_2_ in the absence of antibiotics. Before injections, spirochetes were assessed for motility and quantified by dark-field microscopy (Leica DM2500, Leica Biosystem, Buffalo Grove, IL, USA), recovered by centrifugation at 3000× *g* for 10 min, and resuspended in BSK-H media.

### 4.2. Mouse Skin Biopsy Inoculation, Fixation and Processing

Mouse surplus skin tissues were obtained from Yale University School of Medicine (New Haven, CT, USA). They were from euthanized C57BL/6 mice according to their standard euthanizing protocols from unrelated experiments. The Institutional Animal Care and Use Committee at the University of New Haven deemed that no approval was necessary for the use of (otherwise) discarded tissue. Freshly euthanized mice skins were shaved, bathed in 70% isopropanol, and punched for the retrieval of 3.0 mm skin biopsies (HealthLink, Jacksonville, FL, USA). Intraepidermal injections (10 microliter) with 3/10 mL capacity ultra-fine insulin syringes (BD, Franklin Lakes, NJ, USA) containing a total of 5 × 10^6^ or 1 × 10^7^
*B. burgdorferi* spirochetes in BSK-H media were immediately performed. Infected skin biopsies were maintained in 24-well plates (Corning, Corning, NY, USA) at 33 °C and 5% CO_2_ in either BSK-H containing 6% rabbit serum or DMEM media containing 10% calf serum for 14 days. Biopsies were closely monitored daily, and media was replaced every 2 days of incubation.

On day 14 of incubation, skin biopsies were washed three times with 1× phosphate buffered saline (PBS) pH 7.4 (Sigma, St. Louis, MO, USA) immersed in 4% paraformaldehyde (PFA, Sigma, St. Louis, MO, USA) at 4 °C for overnight and then rinsed with 70% ethanol and stored at 4 °C. In parallel, some samples were immersed in OCT. freezing media (Sakura Finetek, Torrance, CA, USA), snap-frozen with liquid nitrogen, and stored at −80 °C. Formalin-fixed and frozen skin biopsies were processed and sectioned (4 μm) at the Yale University Department of Pathology (New Haven, CT, USA).

### 4.3. Immunohistochemistry (IHC)

Frozen skin tissue sections were thawed and immersed in a Coplin jar containing 4% PFA (Sigma, St. Louis, MO, USA) for 20 min at room temperature, followed by 5 times washes with PBS. Frozen tissue sections were subjected to air-drying for 10 min, and excess PBS surrounding the tissue was gently wiped off with Kimwipes.

Paraffin-embedded fixed tissue sections were deparaffinized on a slide warmer for 15 min and subjected to three 5-min xylene washes. Sections were then immersed in two times in 100%, 90%, and 70% ethanol for 5 min, respectively, and placed under slow running tap water for 45 min.

Then, 100 μL of 1:200 goat serum (diluted in PBS) was added onto the fixed tissue sections for 1 h in a humidified chamber at room temperature. Sections were washed five times in PBS followed by 5 times double distilled water (ddH_2_O) washes. Then, 100 μL of 1:500 rabbit polyclonal anti-alginate antibody (diluted in PBS; generously provided by Dr. Gerald Pier, Harvard Medical School, Ref. [81]) was added to tissue sections and incubated in a humidified chamber overnight at 4 °C. The following morning, the primary polyclonal anti-alginate antibody was removed, and sections were rinsed with 1× PBS for 5 times followed by 5 ddH_2_O washes. Then, 100 μL of 1:200 secondary antibody, goat anti-rabbit IgG (H + L), DyLight 594 conjugated, was added to the tissue samples and incubated in a humidified chamber at 4 °C for 1 h. Secondary antibody was removed, and tissue sections were then rinsed with PBS for 5 times for 1 min each followed by 5 ddH_2_O washes. A total of 100 μL of 1:100 dilution in PBS of anti-*B. burgdorferi* polyclonal antibody labeled with fluorescein isothiocyanate (FITC, raised against *B. burgdorferi* whole cell preparation, rabbit purified IgG, PA-1-73005, Thermo Fisher Scientific, Waltham, MA, USA) was added to tissue sections and incubated in a humidified chamber at room temperature for 1 h. Tissue sections were then washed 5 times with PBS followed by 5 ddH_2_O washes. Tissue section slides were then immersed in a Coplin jar containing 0.1% Sudan black stain (Sigma, St. Louis, MO, USA) for 20 min followed by PBS and ddH_2_O washes as above. Tissue sections used for confocal microscopy were also stained with 300 nM of 4′,6-diamidino-2-phenylindole (DAPI, Thermo Fisher Scientific, Waltham, MA, USA) for 5 min at room temperature followed by PBS and ddH_2_O washes as described above. Tissue section slides were then mounted using PermaFluor (ThermoFisher Scientific, Waltham, MA, USA). Stained tissue sections were visualized via a Leica DM2500 fluorescent microscope equipped with dark field and differential interference contrast (DIC). In addition to infected skin tissue samples, uninfected skin tissue sections were also processed similarly for negative control purposes.

### 4.4. RNA Extraction and Reverse Transcriptase—PCR

Frozen skin tissue sections were thawed for 10 min, and excess liquid was removed with Kimwipes. Tissue was scraped off from the tissue slides into a 1.5 mL micro-centrifuge tube. Then, 1 mL of TRIZOL reagent (Thermo Fisher Scientific, Waltham, MA, USA) was added for 5 min at room temperature. Chloroform (0.2 mL) was then added and the mixture was vigorously shaken for 15 s and incubated for 2 min at room temperature. Samples were then centrifuged at 12,000× *g* for 15 min at 4 °C. Post-centrifugation, the RNA was pipetted out of the aqueous phase of the mixture and transferred to a sterile 1.5 mL micro-centrifuge tube. A total of 0.5 mL of isopropanol was then added to the RNA sample, incubated for 10 min at room temperature, and centrifuged at 12,000× *g* for 10 min at 4 °C. Supernatant was removed and washed with 1 mL of 75% ethanol. The sample was then centrifuged at 7500× *g* for 5 min at 4 °C. The supernatant was removed again, and the RNA pellet was re-suspended in 50 µL RNAse-free H_2_O. To eliminate potential genomic DNA contamination, RNA samples were digested with 2U of Dnase I, Rnase free (Thermo Fisher Scientific, Waltham, MA, USA) for 30 min at 37 °C followed by RNA purification with standard phenol/chloroform extraction as suggested by the manufacturer.

The Transcriptor First Strand cDNA Synthesis Kit (Roche, Basel, Switzerland) was used to convert RNA to DNA following manufacturer’s instruction. In total, 5 μL of the yielded cDNA was used for *B. burgdorferi* 16S ribosomal DNA specific PCR as described previously [39]. Briefly, each reaction consisted of 2.5 μL of 10× PCR buffer (Thermo Fisher Scientific, Waltham, MA, USA), 1.5 mM MgCl_2_, 0.2 mM dNTP mix (Thermo Fisher Scientific, Waltham, MA, USA), 0.2 μM of forward primer (5′-CCTGGCTTAGAACTAACG-3′) and reverse primer (5′-CCTACAAAGCTTATTCCTCAT-3′), and 2.5 U of Taq polymerase (Thermo Fisher Scientific, Waltham, MA, USA). The reactions were adjusted with nuclease-free H_2_O to a final volume of 25 μL. The reactions were denatured at 94 °C for 15 min, followed by 40 cycles of 94 °C for 30 s, 50 °C for 30 s, and 72 °C for 1 min, with a final extension at 72 °C for 5 min. The PCR products were analyzed by standard agarose gel-electrophoresis, and PCR products were purified using the QIAquick PCR purification kit (Qiagen) according to the manufacturer’s instructions. Samples were eluted twice in 30 μL, and the eluates from each sample were pooled and sequenced two times in both directions using the primers that generated the products. Sequencing reactions were performed by Eurofins/MGW/Operon (Huntsville, AL, USA).

In addition to infected skin tissue samples, uninfected tissue sections and *Escherichia coli* RNA were also processed for negative control purposes. *B. burgdorferi* B31 RNA from low passage isolates (<6) was processed for positive control purposes.

### 4.5. Confocal Microscopy

The skin tissue sections were first immunostained for *Borrelia* and alginate as described above and then further analyzed with a confocal scanning laser microscope (Leica DMI6000). ImageJ software (ImageJ, U.S. National Institutes of Health, Bethesda, MD, USA, https://imagej.nih.gov/ij/) was used to process the obtained z-stacks to provide a detailed analysis of the spatial distribution of the different antigens (Plugins: Interactive 3D Surface Plot and Volume Viewer).

### 4.6. Atomic Force Microscopy

Once fluorescent images were acquired, tissue sections were scanned utilizing a contact-mode atomic force microscope (Nanosurf Easyscan) equipped with a SHOCONG-10 probe tip (AppNANO, Mountain View, CA, USA). Images were taken by Hamamatsu ORCA Digital Camera. Images were processed using Gwyddion software in order to generate a 3D topographic view of biofilm forms of *B. burgdorferi*.

### 4.7. Statistical Analysis

Statistical analysis was performed using Student’s t-test (Microsoft Excel, Redmond, WA) on the numbers of observed biofilm aggregates found in the skin tissues. Statistical significance was determined based on *p* values < 0.05.

## 5. Conclusions

This study provides several lines of evidence for the successful development of a novel ex vivo murine skin model for *B. burgdorferi* biofilm research that could be utilized for testing antibiotics and antimicrobial agents as well as for studying the host responses after *B. burgdorferi* infections.

## Figures and Tables

**Figure 1 antibiotics-09-00528-f001:**
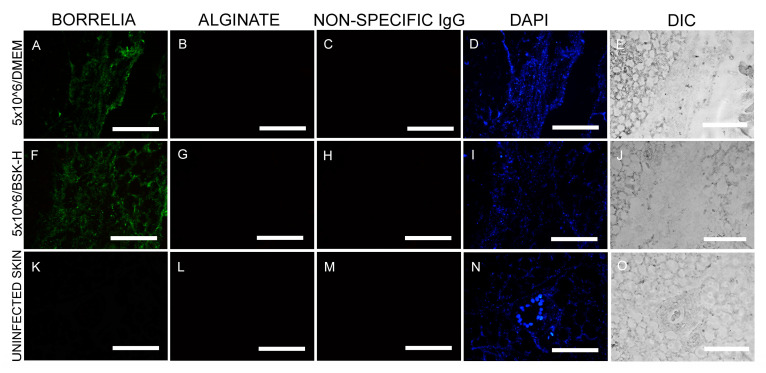
Representative images of immunohistochemical (IHC) experiments for *B. burgdorferi* in murine skin biopsies inoculated at 5 × 10^6^ cells. Panels (**A**,**F**,**K**) show the results for anti-*B. burgdorferi* and panels (**B**,**G**,**L**) for anti-alginate antibodies. DAPI nuclear stains (blue staining: panels (**D**,**I**,**N**)) and differential interference contrast microscopy (panels (**E**,**J**,**O**)) were used to visualize the skin biopsy tissues. Negative controls include staining of uninfected skin (panels (**K**,**L**)) and the use of a non-specific IgG (panels (**C**,**H**,**M**)). 400× magnification, scale bars show 200 μm.

**Figure 2 antibiotics-09-00528-f002:**
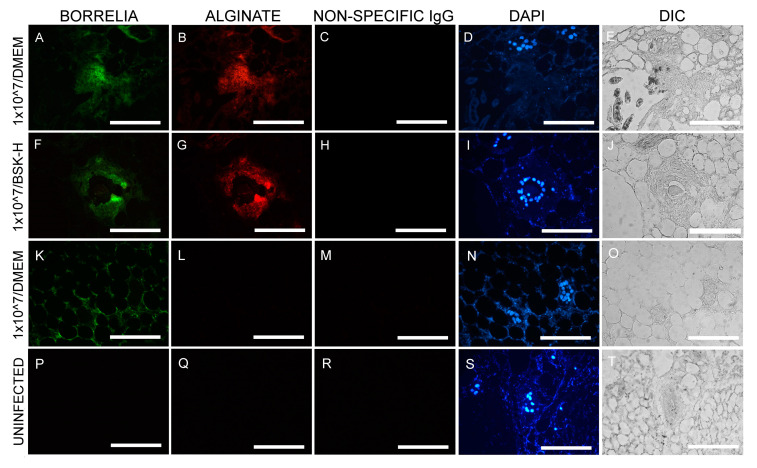
Representative images of IHC experiments for *B. burgdorferi* in murine skin biopsies inoculated at 1 × 10^7^ cells. Panels (**A**,**F**,**K**,**P**) show the results for anti-*B. burgdorferi* and panels (**B**,**G**,**L**,**Q**) for anti-alginate antibodies. DAPI nuclear stains (blue staining: panels **D**,**I**,**N**,**S**)) and differential interference contrast microscopy (panels (**E**,**J**,**O**,**T**)) were used to visualize the tissues. Negative controls include staining of uninfected skin (**P**,**Q**,**R**,**T**) and the use of a non-specific IgG (**C**,**H**,**M**,**R**). 400× magnification, scale bars show 100 μm.

**Figure 3 antibiotics-09-00528-f003:**
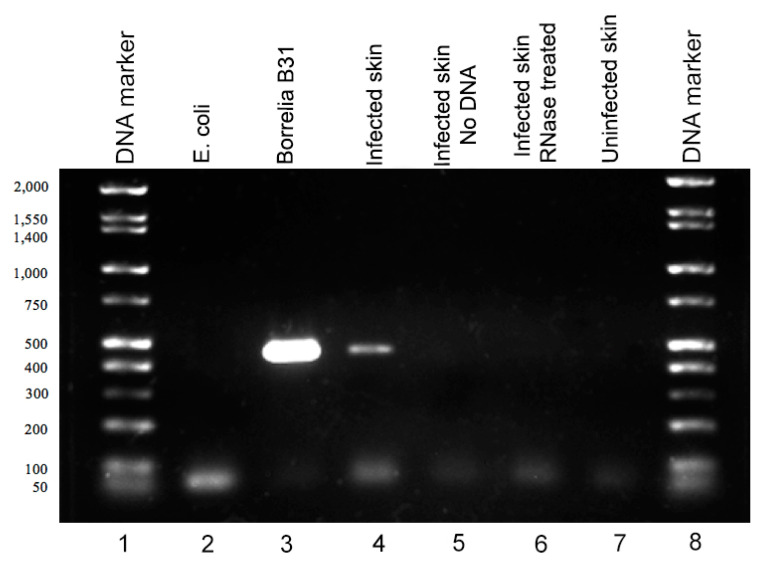
Detection of *B. burgdorferi* RNA in infected murine ex vivo skin biopsies. Gel electrophoresis image of amplified DNA following the RT-PCR protocol. Lane 1 and Lane 8: DNA ladder (Bionexus HI-LO), Lane 2: *E. coli* cDNA template negative control, Lane 3: *B. burgdorferi* B31 cDNA positive control, Lane 4: Borrelia infected murine skin cDNA template, Lane 5: no cDNA template negative control, Lane 6: infected skin with RNase treatment negative control, Lane 7: uninfected murine skin cDNA negative control.

**Figure 4 antibiotics-09-00528-f004:**
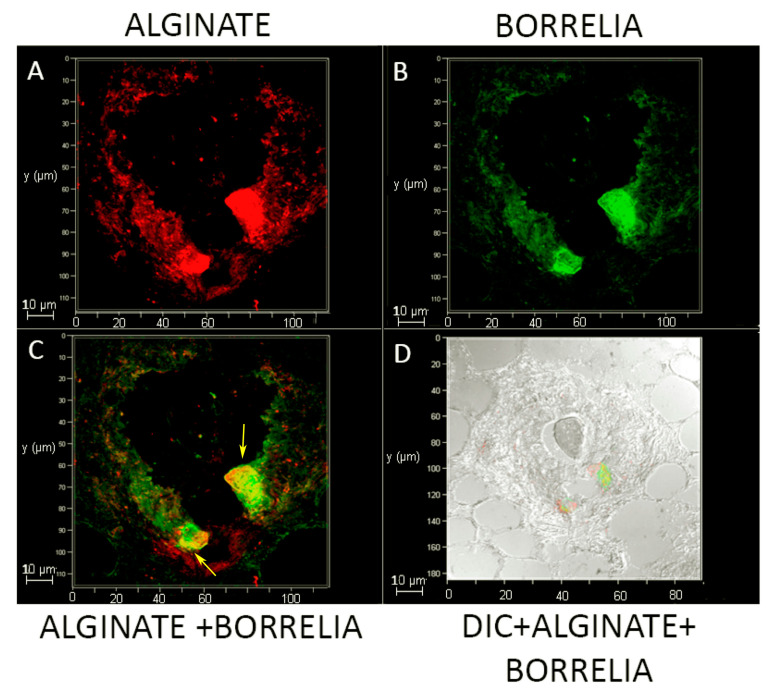
Fluorescent microscopy analysis of biofilm forms of *B. burgdorferi* in infected murine skin biopsies inoculated with 1 × 10^7^ spirochetes and cultured in BSK-H 6% + RS for 14 days. The samples were immunostained for alginate (panel (**A**)) and *B. burgdorferi* (panel (**B**)) antigens. Panel (**C**) shows a merged image with alginate and *B. burgdorferi* staining (yellow arrows demonstrating overlapping staining). Panel (**D**) shows a merged image with differential interference microscopy (DIC) and alginate and *B. burgdorferi* staining; 630× magnification. Scale bars show 10 μm.

**Figure 5 antibiotics-09-00528-f005:**
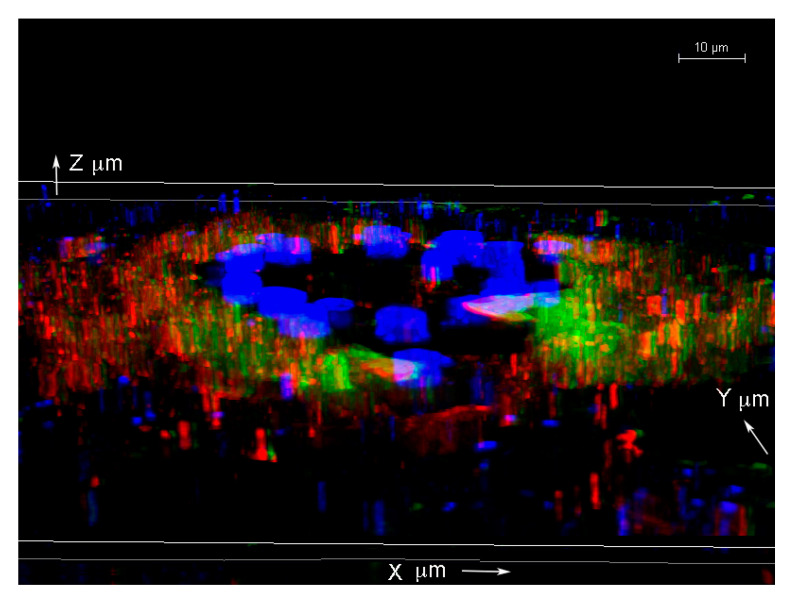
Three-dimensional analysis of biofilm form of *B. burgdorferi* via confocal microscopy in an infected biopsy tissue inoculated with 1 × 10^7^ spirochetes and cultured in BSK-H 6% RS for 14 days. Confocal microscopy analyses of the tissue section performed using merged individual z-stacks to form a composite 3D image from three confocal microscopy channels alginate (red staining), *B. burgdorferi* (green staining), and DAPI (blue staining); 1000× magnification. Scale bar: 10 μm.

**Figure 6 antibiotics-09-00528-f006:**
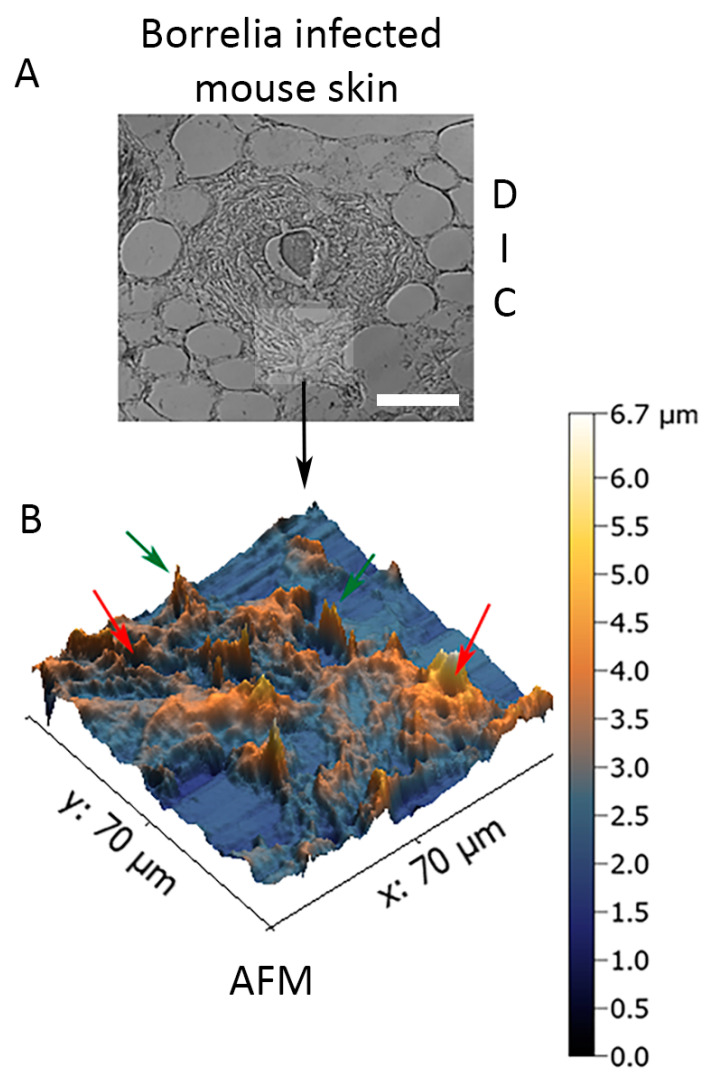
Three-dimensional analysis of biofilm form of *B. burgdorferi* via atomic force microscopy (AFM) in infected biopsies inoculated with 1 × 10^7^ spirochetes and cultured in BSK-H + 6% RS medium for two weeks. Panel (**A**) shows the DIC image of the infected mouse tissue section, which was used for the AFM study (scale bar shows 100 μm). Panel (**B**) shows the results of the AFM analyses. Red arrows show biofilm channels while green arrows show biofilm protrusions.

**Table 1 antibiotics-09-00528-t001:** Quantitative summary of spirochetes found in *B. burgdorferi*-inoculated murine skin biopsies and cultured in BSK-H+6% rabbit serum (RS) and DMEM 10%+calf serum (CS).

Culture Media	Concentration (cells/mL)	Positive Spirochete Slides	# of Spirochetes/mm^2^
BSK-H + 6% RS	5 × 10^6^	30/30	>500
	1 × 10^7^	30/30	>500
DMEM + 10% CS	5 × 10^6^	30/30	>500
	1 × 10^7^	30/30	>500
Uninfected (Both media)	0	0	0

**#** number.

**Table 2 antibiotics-09-00528-t002:** Quantitative summary of biofilms found in *B. burgdorferi*-inoculated murine skin biopsies and cultured in BSK-H + 6% RS and DMEM + 10% CS.

Culture Media	Concentration (cells/mL)	Positive Biofilm Slides	# of Biofilms/mm^2^	Size of Biofilms
BSK-H + 6% RS	5 × 10^6^	0/30	0	0
	1 × 10^7^	12/30	(1–2)	50–300 μm
DMEM + 10% CS	5 × 10^6^	0/30	0	0
	1 × 10^7^	8/30	1	20–200 μm
Uninfected (Both media)	0	0	0	0

**#** number.

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
