# Peer review of "Ex Vivo Murine Skin Model for B. burgdorferi Biofilm"

_antibiotics, 2020, doi:10.3390/antibiotics9090528_

Round 1

Reviewer 1 Report

Revision of manuscript antibiotics-830166

Dear Authors,

Your manuscript entitled “Ex vivo Murine Skin Model for B. burgdorferi Biofilm” reports an interesting experimentation on the development of an ex vivo method to study B. burgdorferi infection. The proposed method was evaluated with different approaches to verify and validate the technique. Overall, the work is well planned and conducted; results were well presented and well and fully discussed.

Below some minor question/corrections:

  • It is very hard to understand results obtained with “1x107 spirochetes”; if I well understood, some biopsies scored negative for alginate staining (and consequentially for biofilm production) as in “1x105 spirochetes” experiment, but some others scored positive for alginate staining, showing biofilm production. The text is not clear and the presence of the 2 images (Figures 2 and 3) does not help, only the Table 2 is explicative about this; please, if it is possible, correct this part.
  • Section “3 Reverse-transcriptase PCR (RT-PCR) analysis on infected murine skin biopsies”: please, correct “2.3”.
  • Lines 371 and 373: Check the text, “µ” is missing;
  • Line 372-373: “The sizes of B. burgdorferi biofilms in ex vivo human skin explants studied here 372 were in a very similar range sized from 20-300 m”, please, check this sentence, maybe it is “mouse” and not “human”.
  • Line 459: How did the Authors estimated the exact number of spirochetes?

I sincerely hope that these suggestions will enhance this manuscript. However, if I have made any errors or misinterpretations, I apologize in advance.

Sincerely

The Reviewer

Author Response

We would like to thank our reviewer for the detailed and constructive review of our manuscript. We have incorporated the suggested revisions and updated the manuscript to address the comments/suggestions made by our reviewer. We have made every attempt to address all concerns and suggestions and make this manuscript ready for publication.

The feedback can be found below in red.

  • It is very hard to understand results obtained with “1x107 spirochetes”; if I well understood, some biopsies scored negative for alginate staining (and consequentially for biofilm production) as in “1x105 spirochetes” experiment, but some others scored positive for alginate staining, showing biofilm production. The text is not clear and the presence of the 2 images (Figures 2 and 3) does not help, only the Table 2 is explicative about this; please, if it is possible, correct this part.

We agree with our reviewer that the separation of the data on spirochetes and biofilm forms into two different figures (Figure 2 and 3) made the results very confusing. Therefore, we merged Figure 2 and 3 to one image (new Figure 2). It is now showing spirochete and biofilm representative structures for the experiments with injections of 1x107 spirochetes. The new figure illustrating the part of the tissue with biofilm formation (alginate staining) and part of the tissue which only has spirochetes (no alginate staining). We hope that new figure will help the reader to better understand the obtained results.

  • Section “3 Reverse-transcriptase PCR (RT-PCR) analysis on infected murine skin biopsies”: please, correct “2.3”. Thank you for finding the typographical error, it is corrected in the revised manuscript.  
  • Lines 371 and 373: Check the text, “µ” is missing; Thank you for finding the font issue, it is corrected in the revised manuscript. 
  • Line 372-373: “The sizes of B. burgdorferi biofilms in ex vivo human skin explants studied here 372 were in a very similar range sized from 20-300 m”, please, check this sentence, maybe it is “mouse” and not “human”. Thank you again, yes it was a mistake and it is corrected.
  • Line 459: How did the Authors estimated the exact number of spirochetes? We determined the number of B. burgdorferi spirochetes on each slide by fluorescent microscopy (× 400 magnification). 

Reviewer 2 Report

The authors report an ex vivo mouse skin model for that leads to the generation of B. burgdoferi biofilm.  The establishment of an ex vivo system to study how B. burgdorferi interacts with and establishes infection within the skin would be an important advance.  Hover, there are a number of concerns about the current study that deserve attention.

  1. The authors state that each mouse 3mm skin biopsy was injected with 10ul of solution containing Bb.  Mouse skin has been reported to be 360mm in thickness including the epidermis (10 μm), corium (250 μm), adipose layer (150 μm).   The calculated volume of a 3mm skin biopsy that is ~0.5mm in height = 3.53ul.  So it is difficult to understand how you inject a tissue of that volume with 10ul?  Perhaps this was misstated.   Also, the location (back, abdomen, etc) the skin biopsy was taken is important and relevant.  When tick naturally feed on mice they usually attach to the ear.
  2. The authors report that injection of 107 Bb lead to the formation of biofilms while 5 x106 did not.  Can the authors comment/discuss on this dosage effect?  Also how does their ex vivo inoculum compare to the inoculum thought to be transmitted by a feeding tick? 
  3. What is the difference between figures 2 and 3 which both report results using an inoculum of 107 In Fig 2, images showing spirochete forms are shown while in figure 3 the author display alginate + biofilms. Are these different focal planes of the same section or are they different specimens, with Fig 2 showing a representative spirochete only sample while figure 3 show a sample where biofilms are found.  This need to be clearly stated. 
  4. Do the alginate + biofilm structure form in vitro in the absence of mouse skin biopsies or do you only see the spirochete form? That control is not presented and is important.
  5. Is there any information on the anti-alginate reagent? There is no reference given, simply who provided it.  Given its importance to the work it should be referenced.
  6. How soon after the biopsy was taken was it placed in culture? At the end of 14 days, was the viability of the cells skin biopsy evaluated? Was the skin anatomy still intact?  This is not obvious from the DIC images as no cellular or anatomical landmarks are identified.   
  7. Was there an attempt to correlate the formation of the biofilms with specific anatomical locations (next to vessels, deep in the dermis, etc). This would be an exciting observation and could be accomplished either using additional markers or using serial sections.
  8. Are the DIC images shown in figures 1-3 the same fields as the adjacent immunofluorescent images? This may be the case for some but several do not seem to have overlapping landmarks.  Overlapping images would be helpful. In the case of figure 2O, the tissue looks degraded.
  9. The authors use a RT-PCR approach to identify Bb rRNA as evidence of the presence of viable bacteria.  I do not believe this result can be used to conclude viability or active transcription (line 344) as an intact nonviable bacterium would also have such a signal.   There are other approaches one might take to determine viability such as the use of re-culture or applying viability dyes.  The authors do not address if the biofilm-like structures contain viable bacteria.
  10. In figures 5 and 6, the authors utilize confocal microscopy to localize the biofilm like structure as “embedded in the mammalian tissue”. As presented the images are not convincing.  This conclusion needs a z-stack 3-D reconstruction which could be provided as a supplemental figure. The image in Figure 6 is 2-D and the authors provide no information to help the reader orientate. Also, since the alginate and Borrelia are thought to overlap, why is there no yellow in the image to indicate that?
  11. Is the AFM image shown in figure 7 from the sample specimen shown in figure 5?  The DIC images look very similar.  How many specimens were analyzed and how many showed these biofilm-like structural features? Also, data should be shown that demonstrates that such structures are not found in 14 day cultures of mouse skin without Bb.

Author Response

We would like to thank our reviewer for the detailed and constructive review of our manuscript. We have incorporated the suggested revisions and updated the manuscript to address the comments/suggestions made by our reviewer. We have made every attempt to address all concerns and suggestions and make this manuscript ready for publication.

The feedback can be found below in red.

  1. The authors state that each mouse 3mm skin biopsy was injected with 10ul of solution containing Bb.  Mouse skin has been reported to be 360mm in thickness including the epidermis (10 μm), corium (250 μm), adipose layer (150 μm).   The calculated volume of a 3mm skin biopsy that is ~0.5mm in height = 3.53ul.  So it is difficult to understand how you inject a tissue of that volume with 10ul?  Perhaps this was misstated.   Also, the location (back, abdomen, etc) the skin biopsy was taken is important and relevant.  When tick naturally feed on mice they usually attach to the ear.The biopsies were taken from the back of the animals. The volume of the tissues was larger than 3.53 microliter because the height of the tissue was about 2 mm which makes the actual volume ~14 microliter. Of course, it means that we probably had some subcutaneous tissues also in the biopsy, however the injection was performed at the epidermis part of the tissues. Yes, the ear and the back of the mouse are the most common parts of tick attachments. Because we needed a relatively high amounts of biopsies, we have chosen the back of the animal for punch biopsies.
  1. The authors report that injection of 107 Bb lead to the formation of biofilms while 5 x106 did not.  Can the authors comment/discuss on this dosage effect?  Also, how does their ex vivo inoculum compare to the inoculum thought to be transmitted by a feeding tick? In our previous in vitro study, we did made a quantitative, comprehensive, in vitro comparison of the biofilms made by Borrelia burgdorferi B31 strain on different biotic and abiotic surfaces (Sapi et al 2012, ref 36) and concluded that the range from 5×106  to 1×107 B31 Borrelia burgdorferi spirochetes are the optimal concentration for seeding and getting significant amounts of biofilm in 7 and 14 days period.Yes, that spirochete concentration we used in this study is above the estimated numbers of spirochetes transmitted by ticks but in a good agreement with the number of the spirochetes present in tick midgut after attachment. Ex vivo organ culture systems are well known to have a limitation in viability especially in long term culture. However, 14 days are generally accepted for ex vivo studies, for example the similar ex vivo study with Borrelia in human tonsillar tissues used 14 days of incubation time. (Duray et al, ref, 55)

  1. What is the difference between figures 2 and 3 which both report results using an inoculum of 107 In Fig 2, images showing spirochete forms are shown while in figure 3 the author display alginate + biofilms. Are these different focal planes of the same section or are they different specimens, with Fig 2 showing a representative spirochete only sample while figure 3 show a sample where biofilms are found.  This need to be clearly stated. We agree with our reviewer that the separation of the 107 inoculation data on spirochetes and biofilm forms into two different figures (Figure 2 and 3) made the results very confusing, therefore we merged Figure 2 and 3 to one new image (new Figure 2) which is now showing spirochetes and biofilm structures for the experiments with 1x107 spirochetes inoculation. We hope that will help the reader to better understand the results. The new figure is showing the part of the tissue with biofilm formation (alginate staining) and part of the tissue which only has spirochetes (no alginate staining).
  2. Do the alginate + biofilm structure form in vitro in the absence of mouse skin biopsies or do you only see the spirochete form? That control is not presented and is important. In our 2012 study, we provided detailed description how Borrelia biofilm can forms and how it depends several factors such as the numbers of spirochetes, the specific culture time and the nature of the surfaces (Sapi et al, 2012, ref 36). As we mentioned above, 5×106 to 1×107 concentrations worked the best for the in vitro experiments up to 14 days culture. Furthermore, for the best biotic surface, collagen was shown to be superior to establish biofilm formation in vitro, that is why we have chosen skin tissues for this study.
  3. Do the alginate + biofilm structure form in vitro in the absence of mouse skin biopsies or do you only see the spirochete form? That control is not presented and is important. We did have several negative controls in our experiment: Uninfected skin tissues stained for both Borrelia and alginate and a non-specific IgG control antibody. Please refer to Figure 1, Panels K-O and Figure 2 Panels P-T.
  4. Is there any information on the anti-alginate reagent? There is no reference given, simply who provided it.  Given its importance to the work it should be referenced. The reference is included, ref #81.
  5. How soon after the biopsy was taken was it placed in culture? At the end of 14 days, was the viability of the cells skin biopsy evaluated? Was the skin anatomy still intact?  This is not obvious from the DIC images as no cellular or anatomical landmarks are identified. The biopsy placed in the culture within 2-3 hours of collection. While the viability of the tissues after 14 days was not evaluated, the integrity of the skin tissues was visualized by DIC microscopy to be sure there are still clear cellular structures.  We, of course, did evaluate the viability and the structural forms of Borrelia. In our future experiments we are planning to evaluate the host tissues as well and do a more detailed analyses of the host tissue environment.
  6. Was there an attempt to correlate the formation of the biofilms with specific anatomical locations (next to vessels, deep in the dermis, etc). This would be an exciting observation and could be accomplished either using additional markers or using serial sections. We agree with the reviewer, that is a very important question and it will be addressed in one of the future experiments. We are planning to analyze the potential differences of locations of spirochetes contra biofilm structures in the skin tissues. In this study, the main goal was to confirm that we can establish biofilm structure is skin ex vivo cultures not just spirochetal structures for futures studies.

  1. The authors use a RT-PCR approach to identify Bb rRNA as evidence of the presence of viable bacteria.  I do not believe this result can be used to conclude viability or active transcription (line 344) as an intact nonviable bacterium would also have such a signal.   There are other approaches one might take to determine viability such as the use of re-culture or applying viability dyes.  The authors do not address if the biofilm-like structures contain viable bacteria. There is a very well reported issue for the difficulties of culturing live Borrelia especially when it is in aggregate forms from infected tissues. Therefore, studies from other research groups used Reverse Transcription PCR method to prove viability of Borrelia in injected tissues (Ref. 21:Embers et al, 2012, Ref 22: Hodzic et al 2014). The idea is based on the fact that bacterial RNA has a very short half-life (2-3 minutes) therefore an active RNA transcription measured by Reverse Transcription PCR can be a proof of viability. Furthermore, to make the methods even more specific for RNA expression, we have used DNase I treatment to eliminate any leftover genomic DNA from the RNA samples. We, however, realized that important experimental step is not described in our manuscript, therefore we added an additional description in our revised Material and Method section – line 544-548.
  2. In figures 5 and 6, the authors utilize confocal microscopy to localize the biofilm like structure as “embedded in the mammalian tissue”. As presented the images are not convincing.  This conclusion needs a z-stack 3-D reconstruction which could be provided as a supplemental figure. We agree that part of the result needs to be better presented with clear labeling, therefore we did provide the additional 3D confocal microscopy image with z-stack analyses for the old Figure 5 (new Figure 4) as Supplemental Figure 1. Old Figure 6 (new Figure 5) was actually a z-stacked confocal image but it was not labeled well so we added additional labels to make it clear the 3D orientations.
  3. The image in Figure 6 is 2-D and the authors provide no information to help the reader orientate.With the additional supplemental figure plus additional sentence on the orientation of the image (lines 265-266) we hope that is now clear to the reader.
  4. Also, since the alginate and Borrelia are thought to overlap, why is there no yellow in the image to indicate that? There is yellow overlap showing on old Figure 5 panel C, now Figure 4 panel C. To make it clearer we added two yellow arrows to help the readers find the overlap.
  5. Is the AFM image shown in figure 7 from the sample specimen shown in figure 5?  The DIC images look very similar.  How many specimens were analyzed and how many showed these biofilm-like structural features? Yes, it is the same image – that was the best biofilm we have found in our experiments (out of 20 total), that is why we have chosen that image to represent both the confocal and AFM analyses. We thought that would help the reader to better understand the biofilm structure if we use the same image and show the analyses with different microscopy methods. We used 30 biopsies/condition - altogether 180 biopsies for all six different type of experiments, which included uninfected biopsies and biopsies infected with different concentrations of Borrelia culture.   We analyzed 30 random sections from each experimental condition, and we have found of total 20 biofilm structures when we inoculated the higher concentration of spirochetes (please see Table 2). Also, data should be shown that demonstrates that such structures are not found in 14 day cultures of mouse skin without Bb. We have only included a statement and the images for the uninfected tissues. However, we agree with the reviewer that data should be clearly stated therefore we included the data in our tables as well (please refer to Table 1 and 2).

Reviewer 3 Report

In this manuscript, the authors develop and studied the ex vivo murine infection model for B. burgdorferi biofilm. The authors used different media combinations in order to find out the most suitable media for growing and maintaining ex vivo biofilm model. They performed various imaging techniques and other molecular analyses to support their hypothesis. All the experiments have been conducted logically and the experimental design is well-suited. Overall, the manuscript is written well. A few corrections are required

Page 8 figure 1 - Please include scale bar for all images

Page 10 figure 2 - Please include scale bar for all images

Page 12 figure 3 - Please include scale bar for all images

Page 15 section 2.4 - since the authors did 3D z-stack analysis of biofilm, it would be useful to include in information on biofilm thickness and roughness information and if the biofilms remained in consistent thickness throughout the entire duration of experiment. This information would be important since thickness and roughness of biofilm can directly alter the treatment regimen when using antibiotics. I would like to see the authors discuss more about this in the discussion section too.

Page 16 figure 5 - what is magnification at which images were taken? Please include that information in figure legend.

Author Response

We would like to thank our reviewer for the detailed and constructive review of our manuscript. We have incorporated the suggested revisions and updated the manuscript to address the comments/suggestions made by our reviewer. We have made every attempt to address all concerns and suggestions and make this manuscript ready for publication.

The feedback can be found below in red.

In this manuscript, the authors develop and studied the ex vivo murine infection model for B. burgdorferi biofilm. The authors used different media combinations in order to find out the most suitable media for growing and maintaining ex vivo biofilm model. They performed various imaging techniques and other molecular analyses to support their hypothesis. All the experiments have been conducted logically and the experimental design is well-suited. Overall, the manuscript is written well. A few corrections are required

Page 8 figure 1 - Please include scale bar for all images – scale bars were included as requested

Page 10 figure 2 - Please include scale bar for all images - scale bars were included as requested

Page 12 figure 3 - Please include scale bar for all images - scale bars were included as requested

Page 15 section 2.4 - since the authors did 3D z-stack analysis of biofilm, it would be useful to include in information on biofilm thickness and roughness information and if the biofilms remained in consistent thickness throughout the entire duration of experiment. This information would be important since thickness and roughness of biofilm can directly alter the treatment regimen when using antibiotics. I would like to see the authors discuss more about this in the discussion section too. We provided a more detailed z-stack analyses for the biofilm structures (supplemental Figure 1 and additional labels to the second confocal image – new Figure 4). The added information included thickness of the biofilm and in discussion we included information whether if that agrees to other in vivo biofilm characteristics (lines 391-393).

Page 16 figure 5 - what is magnification at which images were taken? Please include that information in figure legend. The magnifications in the figure legends of microscopy images were added. Thank you for the suggestion.

Round 2

Reviewer 2 Report

The authors have responded to the comments and have made a number of changes that have improved the manuscript.

However, there is still an issue with the authors interpretation of the RT-PCR results. The author is correct that bacterial RNA does have a short half-life but this generalization applies only to mRNA, not rRNA which tends to be very stable. It would have been very convincing if more than one sample was shown and if other Bb genes were measured and shown to be present.
That said, the authors do mention in the discussion that the PCR results led them to conclude there is active transcription in the tissue (= viable Bb). The authors should also indicate that this is not evidence that the biofilms are actively transcribing as the whole sections analyzed also contain spirochetes and it unknown if the RT-PCR signal is coming from biofilms versus spirochete forms.

Minor issues:
Line 179 – do you mean to refer to Table 2?
Line 205 – do you mean 5 x 106 spirochetes?

Author Response

We would like to thank again our reviewer 2 for the constructive review of our manuscript. We have incorporated the suggested revisions and updated the manuscript to address the comments/suggestions made by our reviewers. We have made every attempt to address all concerns and suggestions and make this manuscript ready for publication.

Reviewer 2

The authors have responded to the comments and have made a number of changes that have improved the manuscript.

However, there is still an issue with the authors interpretation of the RT-PCR results. The author is correct that bacterial RNA does have a short half-life but this generalization applies only to mRNA, not rRNA which tends to be very stable. It would have been very convincing if more than one sample was shown and if other Bb genes were measured and shown to be present.

Yes, we agree that rRNA species are generally more stable then other bacterial RNA species and probably that is the reason we had the best results for the RT-PCR experiments when we used rRNA target. Furthermore, we evaluated the expression of the 16S rRNA species after 2 weeks of culture to avoid the problem of any leftover RNA species in the tissues.

That said, the authors do mention in the discussion that the PCR results led them to conclude there is active transcription in the tissue (= viable Bb). The authors should also indicate that this is not evidence that the biofilms are actively transcribing as the whole sections analyzed also contain spirochetes and it unknown if the RT-PCR signal is coming from biofilms versus spirochete forms.

Thank you, that is correct and we added a sentence to clarify that 16S rRNA expression could be the result of spirochetal gene expression, not necessarily active expression in biofilm structures. Line 362-365.

Minor issues:
Line 179 – do you mean to refer to Table 2? Yes, thank you, it was a typographical error and it corrected.
Line 205 – do you mean 5 x 106 spirochetes? Thank you again, it was a typographical error and it corrected.